# Learning Localized Generative Models for 3D Point Clouds via Graph Convolution

**Diego Valsesia**
Politecnico di Torino
Torino, Italy
diego.valsesia@polito.it

**Giulia Fracastoro**
Politecnico di Torino
Torino, Italy
giulia.fracastoro@polito.it

**Enrico Magli**
Politecnico di Torino
Torino, Italy
enrico.magli@polito.it

## Abstract

Point clouds are an important type of geometric data and have widespread use in computer graphics and vision. However, learning representations for point clouds is particularly challenging due to their nature as being an unordered collection of points irregularly distributed in 3D space. Graph convolution, a generalization of the convolution operation for data defined over graphs, has been recently shown to be very successful at extracting localized features from point clouds in supervised or semi-supervised tasks such as classification or segmentation. This paper studies the unsupervised problem of a *generative* model exploiting graph convolution. We focus on the generator of a GAN and define methods for graph convolution when the graph is not known in advance as it is the very output of the generator. The proposed architecture learns to generate localized features that approximate graph embeddings of the output geometry. We also study the problem of defining an upsampling layer in the graph-convolutional generator, such that it learns to exploit a self-similarity prior on the data distribution to sample more effectively.

## 1 Introduction

Convolutional neural networks are at the core of highly successful models in image generation and understanding. This success is due to the ability of the convolution operation to exploit the principles of locality, stationarity and compositionality that hold true for many data of interest. In particular, feature locality and weight sharing across the data domain greatly reduce the number of parameters in the model, simplifying training and countering overfitting. However, while images are defined on an underlying regular grid structure, several other types of data naturally lie on irregular or non-Euclidean domains (Bronstein et al., 2017). Examples include problems in 3D models (Boscaini et al., 2016; Masci et al., 2015), computational biology (Alipanahi et al., 2015; Duvenaud et al., 2015) or social network graphs (Kipf & Welling, 2016). Defining convolutional architectures on these domains is key to exploit useful priors on the data to obtain more powerful representations.

Graph convolution is emerging as one of the most successful approaches to deal with data where the irregular domain can be represented as a graph. In this case, the data are defined as vectors on the nodes of a graph. Defining a convolution-like operation for this kind of data is not trivial, as even simple notions such as shifts are undefined. The literature has identified two main approaches to define graph convolution, namely spectral or spatial. In the former case (Henaff et al., 2015; Defferrard et al., 2016; Kipf & Welling, 2016), the convolution operator is defined in the spectral domain through the graph Fourier transform (Shuman et al., 2013). Fast polynomial approximations (Defferrard et al., 2016) exist that allow an efficient implementation of the operation. This spectral approach has been successfully used in semi-supervised classification (Kipf & Welling, 2016) and link prediction (Schlichtkrull et al., 2017). However, the main drawback of these techniques is that the structure of the graph is supposed to be fixed and it is not clear how to handle the case where the

graph structure varies. The latter class of methods (Simonovsky & Komodakis, 2017; Wang et al., 2018) defines the convolution operator using a spatial approach by means of local aggregations, i.e., weighted combinations of the vectors restricted to a neighborhood. Since this kind of convolution is defined at a neighborhood level, the operation remains well defined even when the graph varies.

Point clouds are a challenging data type due to the irregular positioning of the points and the fact that a point cloud is an unordered set of points, and therefore any permutation of its members, while changing the representation, does not change its semantic meaning. Some works have addressed supervised problems on point clouds such as classification or segmentation, either through voxelization (Maturana & Scherer, 2015b; Wu et al., 2015), where the irregular point structure is approximated with a regular 3D grid, or by networks like PointNet (Qi et al., 2017a;b) that address the problem of permutation invariance by processing each point identically and independently before applying a globally symmetric operation. The most recent approaches (Simonovsky & Komodakis, 2017; Wang et al., 2018) build graphs in the Euclidean space of the point cloud and use graph convolution operations. This approach has shown multiple advantages in i) reducing the degrees of freedom in the learned models by enforcing some kind of weight sharing, ii) extracting localized features that successfully capture dependencies among neighboring points.

Generative models are powerful tools in unsupervised learning aiming at capturing the data distribution. However, so far little work has been done on generative models for point clouds. Generative models of point clouds can be useful for many tasks that range from data augmentation to shape completion or inpainting partial data thanks to the features learned by the model. Generative Adversarial Networks (GANs) have been shown on images to provide better approximations of the data distribution than variational autoencoders (VAEs) (Larsen et al., 2016), being able to generate sharper images and to capture semantic properties in their latent space. For this reason, it is interesting to study them for unordered point sets. In the first work on the topic, Achlioptas et al. (2017) studied some GAN architectures to generate point clouds. Such architectures use the PointNet approach to deal with the permutation problem at the discriminator and employ a dense generator. However, this means that they are unable to learn localized features or exploit weight sharing.

This paper studies a generative model for point clouds based on graph convolution. In particular, we focus on the GAN generator which is not well explored by the graph convolution literature. This poses a unique challenge: how can one apply a localized operation (the graph convolution) without knowing the domain (the graph) in advance because it is the very output of the generator? We show that the proposed architecture learns domain and features simultaneously and promotes the features to be graph embeddings, i.e. representations in a vector space of the local dependencies between a point and its neighbors. Such localized features learned by the generator provide a flexible and descriptive model. Moreover, we address the problem of upsampling at the generator. While downsampling based on graph coarsening is a staple in (semi-)supervised problems using graph convolution, it is not obvious how to properly upsample the intermediate layers of a graph-convolutional GAN generator. We propose a method exploiting non-local self-similarities in the data distribution.

## 2 METHOD

### 2.1 GRAPH-CONVOLUTIONAL GAN

GANs (Goodfellow et al., 2014) are state-of-the-art generative models composed of a generator and a discriminator network. The generator learns a function mapping a latent vector $\mathbf{z}$ to a sample $\mathbf{x}$ from the data distribution. In the original formulation, the discriminator worked as a classifier trained to separate real samples from generated ones. Recently, the Wasserstein GAN (Arjovsky et al., 2017) addressed the instability and mode collapse issues of the original formulation by modifying the loss function to be a dual formulation of an optimal transport problem using the Wasserstein metric:

$$\min_{G} \max_{\|D\|_L \leq 1} \mathbb{E}_{\mathbf{x} \sim p_{\text{data}}} [D(\mathbf{x})] - \mathbb{E}_{\mathbf{z} \sim p_z} [D(G(\mathbf{z}))] \qquad (1)$$

with a discriminator $D$ and a generator $G$. In this paper, we use the Wasserstein GAN with the gradient penalty method (Gulrajani et al., 2017) to enforce the Lipschitz constraint at the discriminator. In the proposed generative model, we use the Edge-Conditioned Convolution (Simonovsky & Komodakis, 2017) which falls under the category of spatial approaches to graph convolution and is suitable for dealing with multiple arbitrary graphs. Given a layer $l$ with $N^l$ feature vectors $\mathbf{h}_j^l \in \mathbb{R}^{d^l}$ of dimensionality $d^l$, the convolution yields the feature vectors $\mathbf{h}_j^{l+1} \in \mathbb{R}^{d^{l+1}}$ of the next layer by performing, for each node $i$ of the graph, a weighted local aggregation of the feature vectors on

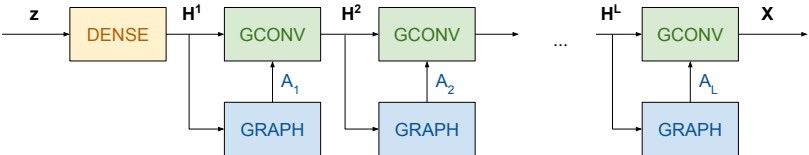

Figure 1: Graph-convolutional generator without upsampling. The feature matrices $\mathbf{H}^l$ are $N \times d^l$, being $N$ the number of points in the point cloud and $d^l$ the number of features at layer $l$. The graph block computes the adjacency matrix of a $k$-nn graph using $\ell_2$ distances between feature vectors.

the neighboring nodes $j \in \mathcal{N}_i^l$, where $\mathcal{N}_i^l$ is the neighborhood of node $i$. The weights of the local aggregation are defined by a fully-connected network $F^l : \mathbb{R}^{d^l} \to \mathbb{R}^{d^l \times d^{l+1}}$, which takes as input the difference between the features of two neighboring nodes and outputs the corresponding weight matrix $\mathbf{\Theta}^{l,ji} = F_{\mathbf{w}^l}^l \left( \mathbf{h}_j^l - \mathbf{h}_i^l \right) \in \mathbb{R}^{d^l \times d^{l+1}}$. Hence, the convolution operation is defined as:

$$\mathbf{h}_i^{l+1} = \sigma \left( \sum_{j \in \mathcal{N}_i^l} \frac{F_{\mathbf{w}^l}^l \left( \mathbf{h}_j^l - \mathbf{h}_i^l \right) \mathbf{h}_j^l}{|\mathcal{N}_i^l|} + \mathbf{h}_i^l \mathbf{W}^l + \mathbf{b}^l \right) = \sigma \left( \underbrace{\sum_{j \in \mathcal{N}_i^l} \frac{\mathbf{\Theta}^{l,ji} \mathbf{h}_j^l}{|\mathcal{N}_i^l|}}_{\text{neighborhood}} + \underbrace{\mathbf{h}_i^l \mathbf{W}^l}_{\text{node}} + \mathbf{b}^l \right) \qquad (2)$$

where $\mathbf{w}^l$ are the weights parametrizing network $F^l$, $\mathbf{W}^l \in \mathbb{R}^{d^l \times d^{l+1}}$ is a linear transformation of the node itself, $\mathbf{b}^l$ a bias, and $\sigma$ a non-linearity. It is important to note that the weights $\mathbf{\Theta}^{l,ij}$ depend only on the difference between the features of the two nodes. This means that two pairs of nodes that have the same difference will have the same weight $\mathbf{\Theta}^{l,ij}$, even if they are in two different regions of the space. This creates weight sharing like in the classical CNNs and represents a strong advantage because it reduces the number of degrees of freedom of the model.

GRAPH-BASED GENERATOR

The focus of this paper is to design a GAN generator that uses localized operations in the form of graphs convolutions. Notice that such operations are able to deal with data in the form of unordered sets, such as points clouds, because they are by design invariant to permutations. However, there are some issues peculiar to the generative problem to be addressed. First, while in supervised problems (Simonovsky & Komodakis, 2017; Wang et al., 2018) or in unsupervised settings involving autoencoders (Yang et al., 2018) the point cloud is known in advance, the intermediate layers of the GAN generator do not know it in advance as it is the very result of the generation operation. It is therefore not obvious how to define an operation that is localized to neighborhoods of a graph that is not known in advance. The solution to this problem is to exploit the pairwise distances ($\|\mathbf{h}_j^{l-1} - \mathbf{h}_i^{l-1}\|$) between node features of the preceding layer to build a $k$-nearest neighbor graph. Figure 1 shows a block diagram of a graph-based generator where each graph convolution block uses the graph constructed from the input features of the block itself. The intuition behind this solution is that this promotes the features to become graph embeddings, i.e. representations in a high-dimensional metric space of relationships between points. Going through the generator network from the latent space towards the point cloud output, these embeddings are assembled hierarchically and their associated graphs represent better and better approximations of the graph of the output point cloud.

According to the definition of graph convolution in (2), the new features of a node are a weighted combination of the features of the node itself and of the neighbors as determined by the graph construction. Notice that this localized approach differs from the one in Achlioptas et al. (2017) where the generator of the r-GAN model is a fully-connected network, therefore unable to provide any localized interpretation of its hidden layers. It also differs from the PointNet (Qi et al., 2017a) and PointNet++ (Qi et al., 2017b) architectures. PointNet processes each point independently with the same weights and then aggregates them using a globally symmetric operation to deal with the permutation invariance problem. PointNet++ extends this work using some localized operations. However, the key difference with the work in this paper is that PointNet and PointNet++ are not generative models, but are used in supervised problems such as classification or segmentation. Other works explore likelihood-based generative models, typically in the form of variational autoencoders (Fan et al., 2017; Nash & Williams, 2017; Litany et al., 2017). The most similar approach to the method of this paper is the one in Grover et al. (2018), with the key difference being that a distribution over adjacency matrices of graphs is learned using a spectral graph-convolutional VAE.

Figure 2: Graph-convolutional generator with upsampling. The feature matrices $\mathbf{H}^l$ have size $N^l \times d^l$ being $N^l$ the number of points at layer $l$ and $d^l$ the number of features at layer $l$. The graph block computes the adjacency matrix of a $k$-nn graph using $\ell_2$ distances between feature vectors. The upsamp block computes $N^{l+1} - N^l$ new points and concatenates them to the input ones.

## 2.2 Upsampling

The previous section presented the basic outline of a graph-based generator in a GAN. However, one evident shortcoming is the fixed number of points throughout the generator, which is determined by the number of output points. Many data of interest typically display some kind of regularity in the form of multi-resolution or other kinds of compositionality whereby points can be predicted from a smaller number of neighboring points. In the case of 2D images, lower resolutions provide a prediction of higher resolutions by supplying the low-frequency content and the upsampling operation is straightforward. In fact, convolutional GANs for image generation are composed of a sequence of upsampling and convolutional layers. Extending upsampling to deal with the generation of sets of points without a total ordering is not a trivial task. Many works have addressed the problem of upsampling 3D point clouds, e.g., by creating grids in the 3D space (Maturana & Scherer, 2015a). Notice, however, that introducing upsampling to interleave the graph-convolutional layers outlined in the previous section is a more complex problem because the high dimensionality of the feature vectors makes the gridding approach unfeasible.

If we consider the $l$-th generator layer, we want to define an upsampling operation that, starting from the graph convolution output $\mathbf{H}^l \in \mathbb{R}^{N^l \times d^l}$, generates $N^l$ new feature vectors $\tilde{\mathbf{H}}^l \in \mathbb{R}^{N^l \times d^l}$. Then, these new feature vectors are concatenated to $\mathbf{H}^l$ in order to obtain the output $\mathbf{H}^{l,\mathrm{up}} \in \mathbb{R}^{2N^l \times d^l}$. We propose to define an upsampling operation using local aggregations. In this case, the upsampling operation becomes similar to a graph convolution. Given a feature vector $\mathbf{h}_i^l \in \mathbb{R}^{d^l}$, we consider a set of neighbors $\mathcal{N}_i^l$ and we define the new feature vector $\tilde{\mathbf{h}}_i^l \in \mathbb{R}^{d_l}$ as follows

$$\tilde{\mathbf{h}}_i^l = \sigma \left( \sum_{j \in \mathcal{N}_i^l} \frac{\mathrm{diag}\left( U_{\tilde{\mathbf{w}}^l}^l \left( \mathbf{h}_j^l - \mathbf{h}_i^l \right) \right) \mathbf{h}_j^l}{|\mathcal{N}_i^l|} + \mathbf{h}_i^l \Gamma^{l,i} + \mathbf{b}^l \right) = \sigma \left( \sum_{j \in \mathcal{N}_i^l} \frac{\Gamma^{l,ji} \mathbf{h}_j^l}{|\mathcal{N}_i^l|} + \mathbf{h}_i^l \Gamma^{l,i} + \mathbf{b}^l \right)$$

where $U^l : \mathbb{R}^{d^l} \to \mathbb{R}^{d^l}$ is a fully-connected network which given the difference between $\mathbf{h}_i^l$ and $\mathbf{h}_j^l$ outputs the weight vector $\boldsymbol{\gamma}^{l,ij} \in \mathbb{R}^{d^l}$, which is used to create the diagonal matrix $\Gamma^{l,ji} = \mathrm{diag}\left( \boldsymbol{\gamma}^{l,ji} \right)$. $\tilde{\mathbf{w}}^l$ and $\mathbf{b}^l$ are model parameters that are updated only during training. It is important to note that, differently from the graph convolution described in 2.1 where $\Theta^{l,ij}$ and $\tilde{\mathbf{W}}^l$ are dense matrices, in this case we use diagonal matrices. This means that during the upsampling operation the local aggregation treats each feature independently. This also reduces the number of parameters.

## 2.3 Graph embedding interpretation

Graph embeddings (Goyal & Ferrara, 2017) are representations of graphs in a vector space where a feature vector is associated to each node of the graph. For what concerns this paper we consider the following definition of graph embedding, focused on predicting edges from the feature vectors.

**Definition 1** *Given a graph $\mathcal{G} = (\mathcal{V}, \mathcal{E})$, a graph embedding is a mapping $f : i \to \mathbf{h}_i \in \mathbb{R}^d, \forall i \in \mathcal{V}$, such that $d \ll |\mathcal{V}|$ and the function $f$ is defined such that if we consider two pairs of nodes $(i, j)$ and $(i, k)$ where $(i, j) \in \mathcal{E}$ and $(i, k) \notin \mathcal{E}$ then $\|\mathbf{h}_i - \mathbf{h}_j\| < \|\mathbf{h}_i - \mathbf{h}_k\|$.*

The graph-convolutional generator presented in this paper can be interpreted as generating graph embeddings of the nearest-neighbor graph of the output point cloud at each hidden layer, thus creating features that are able to capture some properties of the local topology. In order to see why this is the case, we analyze the architecture in Fig. 1 backwards from the output to the input. The final output $\mathbf{x}$ is the result of a graph convolution aggregating features localized to the nearest-neighbor

Table 1: Generator architecture

| No upsampling | | Upsampling | |
| --- | --- | --- | --- |
| **Layer** | **Output size** | **Layer** | **Output size** |
| Latent | $1 \times 128$ | Latent | $1 \times 128$ |
| Dense | $2048 \times 32$ | Dense | $128 \times 96$ |
| Gconv | $2048 \times 32$ | Gconv | $128 \times 48$ |
| | | Upsamp | $256 \times 48$ |
| Gconv | $2048 \times 24$ | Gconv | $256 \times 32$ |
| | | Upsamp | $512 \times 32$ |
| Gconv | $2048 \times 16$ | Gconv | $512 \times 16$ |
| | | Upsamp | $1024 \times 16$ |
| Gconv | $2048 \times 8$ | Gconv | $1024 \times 8$ |
| | | Upsamp | $2048 \times 8$ |
| Gconv | $2048 \times 3$ | Gconv | $2048 \times 3$ |

graph computed from the features of the preceding layer. Since the GAN objective is to match the distribution of the output with that of real data, the neighborhoods identified by the last graph must be a good approximation of the neighborhoods in the true data. Therefore, we say that features $\mathbf{H}^L$ are a graph embedding in the sense that they allow to predict the edges of the output graph from their pairwise distances. Proceeding backwards, there is a hierarchy of graph embeddings as the other graphs are constructed from higher-order features.

Notice that the upsampling operation in the architecture of Fig. 2 affects this chain of embeddings by introducing new points. While the graph convolution operation promotes the features of all the points after upsampling to be graph embeddings, the upsampling operation affects which points are generated. In the experiments we show that the upsampling method approximately maintains the neighborhood shape but copies it elsewhere in the point cloud. This suggests a generation mechanism exploiting self-similarities between the features of the point cloud at different locations.

## 3 EXPERIMENTS

We tested the proposed architecture by using three classes of point clouds taken from the ShapeNet repository (Chang et al., 2015): "chair", "airplane" and "sofa". A class-specific model is trained for the desired class of point clouds. Since the focus of this paper is the features learned by the generator, the architecture for the discriminator is the same as the one of the r-GAN in Achlioptas et al. (2017), with 4 layers with weights shared across points (number of output features: 64, 128, 256, 512) followed by a global maxpool and by 3 dense layers. The generator architecture is reported in Table 1. The graph is built by selecting the 20 nearest neighbors in terms of Euclidean distance in the feature space. We use Leaky ReLUs as nonlinearities and RMSProp as optimization method with a learning rate equal to $10^{-4}$ for both generator and discriminator. Batch normalization follows every graph convolution. The gradient penalty parameter of the WGAN is 1 and the discriminator is optimized for 5 iterations for each generator step. The models have been trained for 1000 epochs. For the "chair" class this required about 5 days without upsampling and 4 days with upsampling.

### 3.1 GENERATED POINT CLOUD ASSESSMENT

In this section we perform qualitative and quantitative comparisons with the generated point clouds.

#### VISUAL RESULTS

We first visually inspect the generated point clouds from the classes "chair" and "airplane", as shown in Fig. 3. The results are convincing from a visual standpoint and the variety of the generated objects is high, suggesting no mode collapse in the training process. The distribution of points on the object is quite uniform, especially for the method with upsampling.

#### COMPARISONS WITH STATE-OF-THE-ART

To the best of our knowledge this is the first work addressing GANs for point clouds learning localized features. We compare the proposed GAN for point cloud generation with other GANs able to

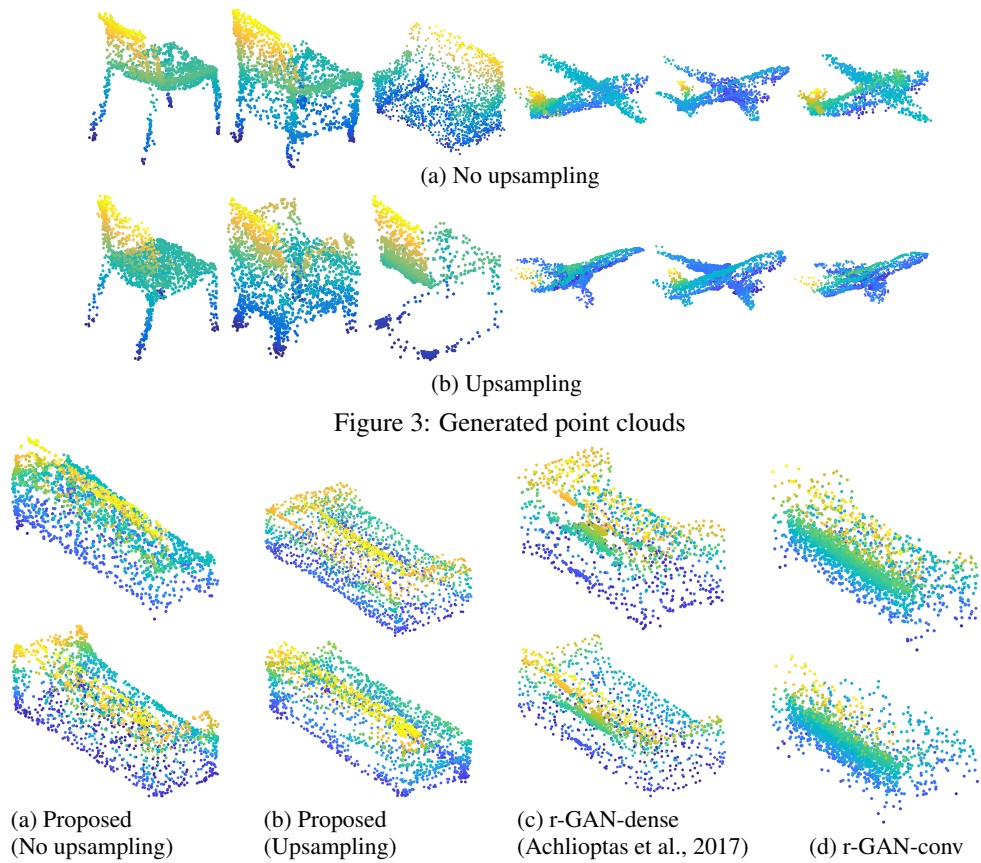

(a) No upsampling

(b) Upsampling

Figure 3: Generated point clouds

(a) Proposed
(No upsampling)

(b) Proposed
(Upsampling)

(c) r-GAN-dense
(Achlioptas et al., 2017)

(d) r-GAN-conv

Figure 4: Generated point clouds from different methods

deal with unordered sets of points. In particular, the "r-GAN-dense" architecture in Achlioptas et al. (2017) has a dense generator, which is unable to generate localized representations because there is no mapping between points and feature vectors. As an additional baseline variant, dubbed "r-GAN-conv", we study the use of a generator having as many feature vectors as the points in the point cloud and using a size-1 convolution across the points. Notice that the graph convolution we use can be seen as a generalization of this model, aggregating the features of neighboring points instead of processing each point independently. We point out that we cannot compare the proposed method in a fair way with the variational autoencoders mentioned in Sec. 2.1: Fan et al. (2017) generate point clouds conditioned on an input image; Nash & Williams (2017) use object segmentation labels to generate point clouds by parts; Litany et al. (2017) focus on generating vertices on meshes with a fixed and given topology.

In order to perform a quantitative evaluation of the generated point clouds we use the evaluation metrics proposed in Achlioptas et al. (2017), employing three different metrics to compare a set of generated samples with the test set. The first one is the Jensen-Shannon divergence (JSD) between marginal distributions defined in the 3D space. Then, we also evaluate the coverage (COV) and the minimum matching distance (MMD), as defined in Achlioptas et al. (2017), using two different point-set distances, the earth mover's distance (EMD) and the Chamfer distance (CD). Table 2 shows the obtained results. As can be seen, the proposed methods achieve better values for the metrics under consideration. In particular, the method with upsampling operations is consistently the better. Notice that Achlioptas et al. (2017) report that the Chamfer distance is often unreliable as it fails to penalize non-uniform distributions of points. Fig. 4 visually shows that the proposed methods generate point clouds with better-distributed points, confirming the quantitative results. In particular, the r-GAN-dense shows clusters of points, while the r-GAN-conv also exhibits noisy shapes.

## 3.2 GENERATOR FEATURE ANALYSIS

In this section we quantitatively study the properties of the features in the layers of the generator.

Table 2: Quantitative comparisons

| Class | Model | JSD | MMD-CD | MMD-EMD | COV-CD | COV-EMD |
|-------|-------|-----|--------|---------|--------|---------|
| Chair | **r-GAN-dense** | 0.238 | **0.0029** | 0.136 | **33** | 13 |
| | **r-GAN-conv** | 0.517 | 0.0030 | 0.223 | 23 | 4 |
| | **Prop. (no up.)** | 0.119 | 0.0033 | 0.104 | 26 | 20 |
| | **Prop. (up.)** | **0.100** | **0.0029** | **0.097** | 30 | **26** |
| Sofa | **r-GAN-dense** | 0.221 | **0.0020** | 0.146 | 32 | 12 |
| | **r-GAN-conv** | 0.293 | 0.0025 | 0.110 | 21 | 12 |
| | **Prop. (no up.)** | 0.095 | 0.0024 | 0.094 | 25 | 19 |
| | **Prop. (up.)** | **0.063** | **0.0020** | **0.083** | **39** | **24** |

Figure 6: No upsampling: $k$-means clustering of features of intermediate layers, highlighted onto the output point cloud (leftmost: output of dense layer, rightmost: output point cloud). Notice how layer features generate clusters that are progressively more localized in the output geometry.

GRAPH EMBEDDING AND FEATURE RADIUS WITHOUT UPSAMPLING

Referring to Table 1, the output of each layer is a matrix where every point is associated to a feature vector. In Sec. 2.3 we claimed that these features learned by the generator are graph embeddings.

We tested this hypothesis by measuring how much the adjacency matrix of the final point cloud, constructed as a nearest-neighbor graph in 3D, is successfully predicted by the nearest-neighbor adjacency matrix computed from hidden features. This is shown in Fig.5 which reports the percentage of edges correctly predicted as function of the number of neighbors considered for the graph of the output point cloud and a fixed number of 20 neighbors in the feature space. Notice that layers closer to the output correctly predict a higher percentage of edges and in this sense are better graph embeddings of the output geometry.

Fig. 6 shows another experiment concerning localization of features. We applied $k$-means with 6 clusters to the features of intermediate layers and represented the cluster assignments onto the final point cloud. This experiment confirms that the features are highly localized and progressively more so in the layers closer to the output.

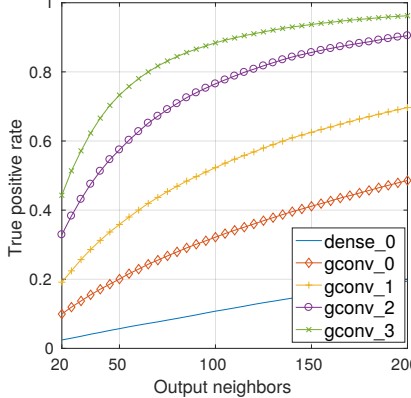

Figure 5: Accuracy of edge prediction from intermediate layer features.

We further investigated the effective receptive field of the convolution operation in Fig. 7a. This figure reports histograms of Euclidean distances measured on the output point cloud between neighbors as determined by the nearest neighbor graph in one of the intermediate layers. We can see that layers closer to the output aggregate points which are very close in the final point cloud, thus implementing a highly localized operation. Conversely, layers close to the latent space perform more global operations.

UPSAMPLING RESULTS

The main drawback of the model without upsampling is the unnecessarily large number of parameters in the first dense layer. This is solved by the introduction of the upsampling layers which aim at

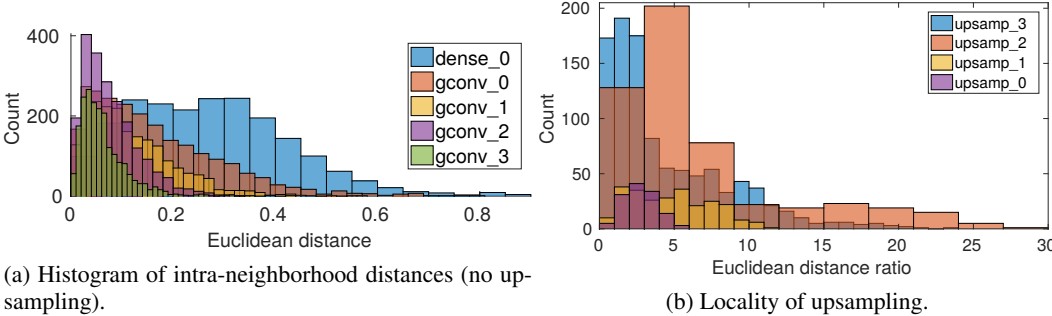

(a) Histogram of intra-neighborhood distances (no up-sampling).

(b) Locality of upsampling.

Figure 7: (a) Neighborhoods are computed as 20-nearest neighbors in the feature space of each layer. Distances in abscissa are distances in the 3D point cloud. The layer features create neighborhoods that are progressively more localized with respect to the output geometry. (b) Upsampled points are non-local in the output geometry.

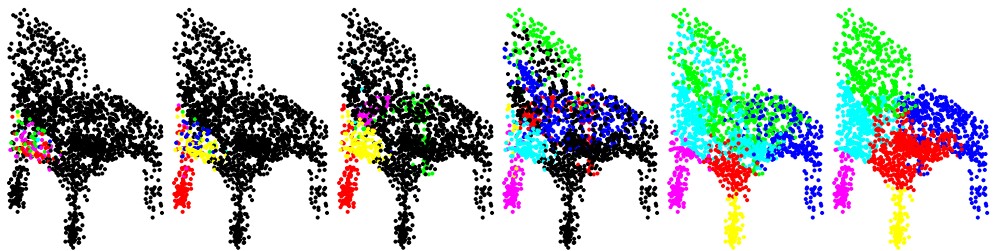

Figure 8: Aggregation upsampling: $k$-means clustering of features of intermediate layers after up-sampling, highlighted onto the output point cloud (leftmost: output of dense layer, rightmost: output point cloud). Black points are not yet generated at the intermediate layers.

exploiting hierarchical priors to lower the number of parameters by starting with a lower number of points and progressively predicting new points from the generated features.

The proposed upsampling technique based on local aggregations computes a new point as a weighted aggregation of neighboring points. The weights of the aggregation are learned by the network, thus letting the network decide the best method to create a new point from a neighborhood, at the expense of an increased number of total parameters. The experiment in Figs. 7b and 8 shows an interesting behavior. First, the generated points are not close to the original point: Fig. 7b shows the ratio between the generator-generated distance and the average neighborhood distance (neighborhoods are defined in the feature space, while distances are measured as Euclidean distances on the output 3D point cloud) and since it is usually significantly larger than 1, we can conclude that the generated point is far from the original generating neighborhood. Then, the clusters in Fig. 8 show that the points in the first layers are not uniformly distributed over the point cloud, but rather form parts of it. The mechanism learned by the network to generate new points is essentially to apply some mild transformation to a neighborhood and copy it in a different area of the feature space. The generated points will no longer be close to their generators, but the structure of the neighborhood resembles the one of the generating neighborhood. This notion is similar to the second-order proximity in the graph embedding literature (Goyal & Ferrara, 2017) and it seems that this operation is exploiting the inherent self-similarities between the data features at distant points. To validate this hypothesis we measured two relevant quantities. First, we considered a point $i$, its neighbors $\mathcal{N}_i^l$ before upsampling, their corresponding points generated by the upsampling operation $\{i^{\mathrm{up}}, \mathcal{N}_i^{l,\mathrm{up}}\}$ and the neighborhood $\mathcal{N}_{i^{\mathrm{up}}}^l$ of point $i^{\mathrm{up}}$. We measured the average percentage of points in $\mathcal{N}_i^{l,\mathrm{up}}$ that were generated from points in $\mathcal{N}_i^l$, i.e. $|\mathcal{N}_{i^{\mathrm{up}}}^l \cap \mathcal{N}_i^{l,\mathrm{up}}|/|\mathcal{N}_i^l|$, as reported in Table 3. The result shows that the neighborhood of a generated point is almost entirely generated by the points that were neighbors of the generator, and that the new points are not neighbors of the original ones. This behavior is consistent over different layers. Then, we measured the Euclidean distances in the feature space between point $i$ and its neighbors $\mathcal{N}_i^l$ and between point $i^{\mathrm{up}}$ and $\mathcal{N}_i^{l,\mathrm{up}}$. Table 3 reports the correlation coefficient between those distance vectors, which suggests that the shape of the neighborhood is fairly conserved.

Table 3: Upsampling self-similarity

|  | upsamp_0 | upsamp_1 | upsamp_2 | upsamp_3 |
|---|---|---|---|---|
| **Percentage of neighbors** | $(71.8 \pm 7.6)\,\%$ | $(69.6 \pm 2.8)\,\%$ | $(61.6 \pm 3.3)\,\%$ | $(66.4 \pm 3.5)\,\%$ |
| **Distance correlation** | $0.56 \pm 0.15$ | $0.60 \pm 0.03$ | $0.53 \pm 0.03$ | $0.61 \pm 0.04$ |

## 4 CONCLUSIONS

We presented a GAN using graph convolutional layers to generate 3D point clouds. In particular, we showed how constructing nearest neighbor graphs from generator features to implement the graph convolution operation promotes the features to be localized and to approximate a graph embedding of the output geometry. We also proposed an upsampling scheme for the generator that exploits self-similarities in the samples to be generated. The main drawback of the current method is the rather high complexity of the graph convolution operation. Future work will focus on reducing the overall complexity, e.g., in the graph construction operation, and study new upsampling schemes.

## 5 ACKNOWLEDGEMENTS

The research leading to these results has been partially funded by the SmartData@PoliTO center for Big Data and Machine Learning technologies. The research leading to this publication has received funding from Regione Piemonte under research grant "DISLOMAN: dynamic integrated shopfloor operation management for Industry 4.0".

We thank Nvidia for donating a Quadro P6000 GPU for this work.

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

# 6 APPENDIX

| Class | Model | JSD | MMD-CD | MMD-EMD | COV-CD | COV-EMD |
|---|---|---|---|---|---|---|
| Chair | **r-GAN-dense** | 0.238 | **0.0029** | 0.136 | **33** | 13 |
| | **r-GAN-conv** | 0.517 | 0.0030 | 0.223 | 23 | 4 |
| | **Prop. (no up.)** | 0.119 | 0.0033 | 0.104 | 26 | 20 |
| | **Prop. (up.)** | **0.100** | **0.0029** | **0.097** | 30 | **26** |
| Airplane | **r-GAN-dense** | 0.182 | 0.0009 | 0.094 | **31** | 9 |
| | **r-GAN-conv** | 0.350 | **0.0008** | 0.101 | 26 | 7 |
| | **Prop. (no up.)** | 0.164 | 0.0010 | 0.102 | 24 | 13 |
| | **Prop. (up.)** | **0.083** | **0.0008** | **0.071** | **31** | **14** |
| Sofa | **r-GAN-dense** | 0.221 | **0.0020** | 0.146 | 32 | 12 |
| | **r-GAN-conv** | 0.293 | 0.0025 | 0.110 | 21 | 12 |
| | **Prop. (no up.)** | 0.095 | 0.0024 | 0.094 | 25 | 19 |
| | **Prop. (up.)** | **0.063** | **0.0020** | **0.083** | **39** | **24** |

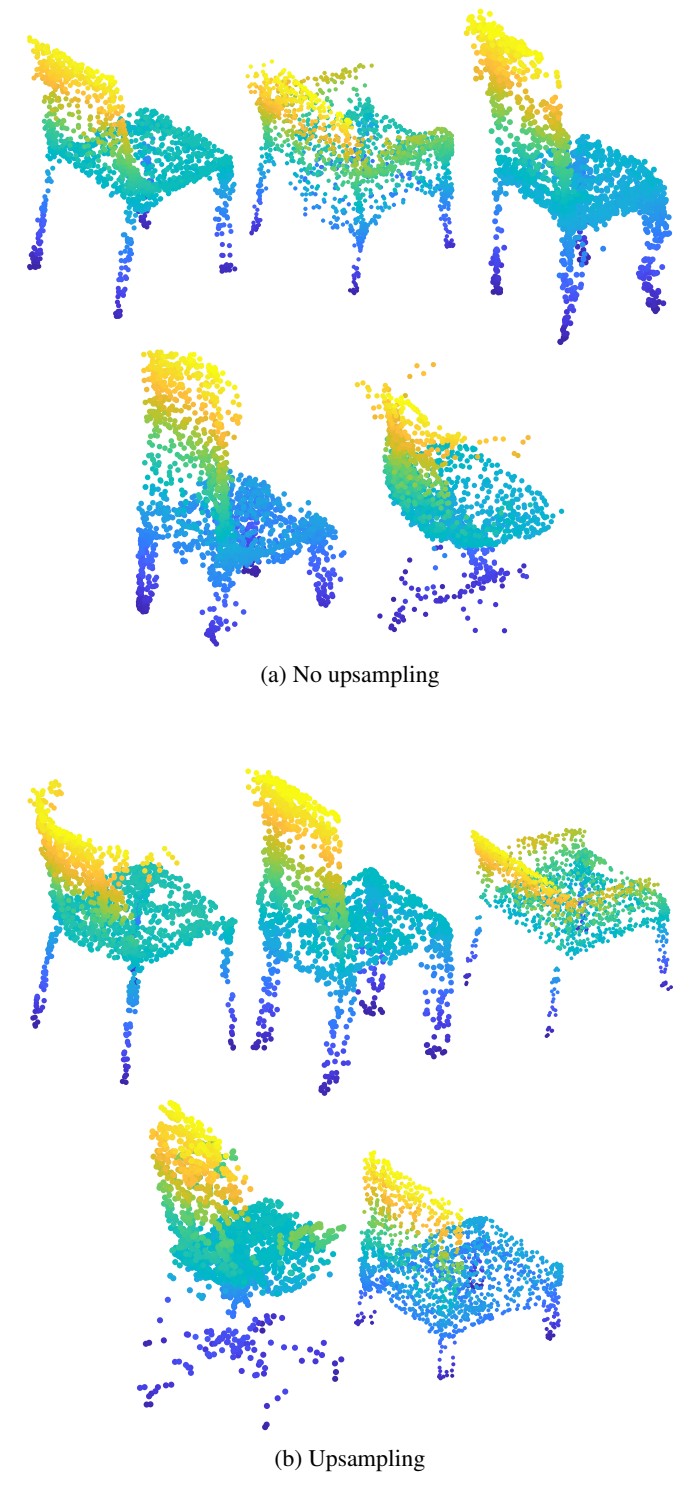

(a) No upsampling

(b) Upsampling

Figure 9: Generated point clouds for chair class

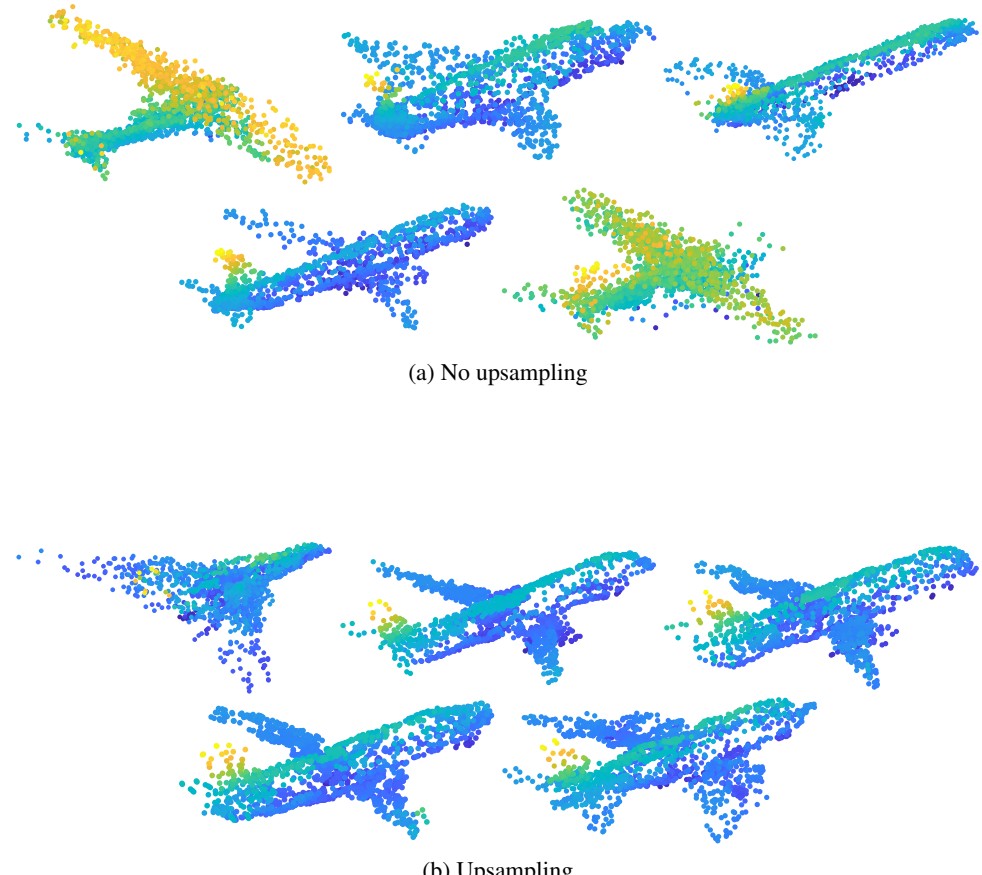

(a) No upsampling

(b) Upsampling

Figure 10: Generated point clouds for airplane class

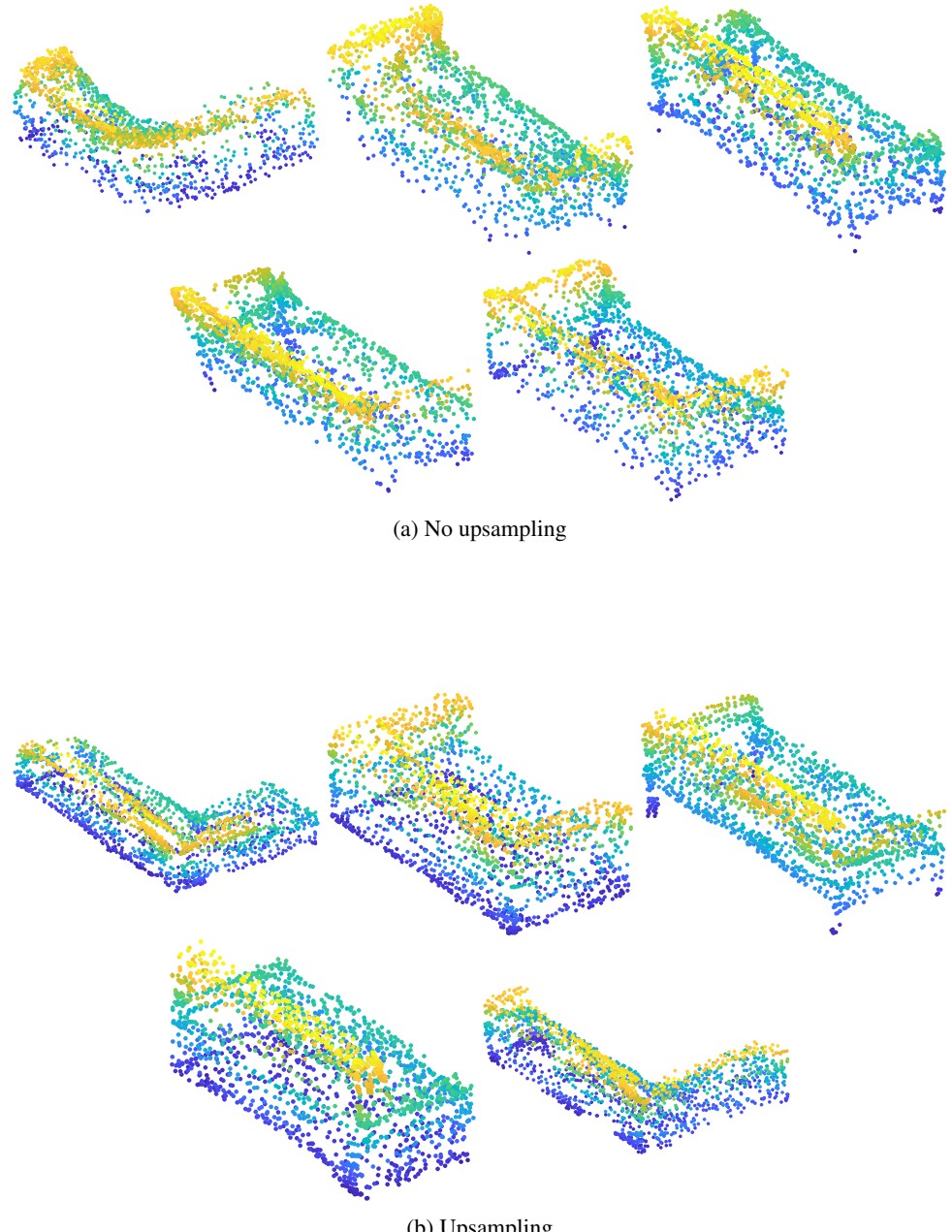

(a) No upsampling

(b) Upsampling

Figure 11: Generated point clouds for sofa class

