# OpenReview forum: "Learning Localized Generative Models for 3D Point Clouds via Graph Convolution"
_ICLR.cc/2019/Conference_

### Official Review · AnonReviewer2 · 2018-10-29
**Nice contribution within the focus of ICLR**

**Rating:** 7
**Confidence:** 3

**Review:**

The authors present a method for generating points clouds with the help of graph convolution and a novel upsampling scheme. The proposed method exploits the pairwise distances between node features to build a NN-graph. The upsampling scheme generates new points via a slimmed down graph convolution, which are then concatenated to the initial node features. The proposed method is evaluated on four categories of the ShapeNet dataset. Resulting point clouds are evaluated via a qualitative and quantitative comparison to r-GAN.

As far as I know, the paper introduces an overall novel and interesting idea to generate point clouds with localized operations.


The following questions could be addressed by the authors in a revised manuscript:

* The upsampling operation is not well motivated, e.g., neighboring node features are weighted independently, but root node features are not. What is the intuition besides reducing the number of parameters? Are there significant differences when not using diagonal weight matrices?
* As computation of pairwise node feature distances and graph generation based on nearest neighbors are expensive tasks, more details on the practical running time and theoretical complexity should be provided. Can the complexity be reduced by rebuilding graphs only after upsampling layers? How would this impact the performance of the proposed model?
* Although the evaluation on four categories is reported, Table 2 only gives results for two categories.
* How is the method related to GANs which generates graphs, such as GraphGAN or NetGAN?

---

> ### Author Response · Authors · 2018-11-14
> **Thank you for your comments**
>
> We thank the reviewer for the comments.
>
> *"The upsampling operation is not well motivated, e.g., neighboring node features are weighted independently, but root node features are not. What is the intuition besides reducing the number of parameters? Are there significant differences when not using diagonal weight matrices?":
> We thank the reviewer for spotting this issue. There is a typo in the manuscript. The root node is not weighted by a dense matrix but by a diagonal matrix to be consistent with the operation on its neighbors. Anyway, we also experimented with dense matrices and using independent features performed slightly better. The intuition behind this definition of upsampling is that we want to generate a new point in the same space of the original points (hence not mixing the features) with scaling and translation operations. Notice that if we are generating two points starting from two root points sharing the same neighborhoord, the operation will perform something like adding a "residual" to the root point that depends on the neighborhood and since both points share the neighborhood this residual will be similar, thus approximately preserving the structure of the neighborhood in the position in the space. This allows to exploit the self-similarity prior as confirmed by the experiments.
>
>
> * "As computation of pairwise node feature distances and graph generation based on nearest neighbors are expensive tasks, more details on the practical running time and theoretical complexity should be provided. Can the complexity be reduced by rebuilding graphs only after upsampling layers? How would this impact the performance of the proposed model?":
> As explained in the response to reviewer 1, complexity is one of the main issues of the proposed method. The computation of pairwise distances at each layer contributes to increase the complexity with respect to classical convolutional methods. However, we already partially addressed it in two ways: 1) by developing an upsampling scheme, which allows to reduce the number of points in the first layers; 2) the graph constructed from the upsampled output can already reuse the distances computed from the input and only computes the ones with the new points. We have added a comment on the complexity and the running time to train the model for 1000 epochs in the revised paper.
> We have not investigated using a single graph for both convolution and upsampling as suggested by the reviewer since every graph convolution operation should provide a better graph embedding at its output, thereby improving the perfomance of the upsampling layer. The dynamic update of the graph at every layer is what enables the network to learn graph embeddings, so modifications to this architecture should be studied carefully. One option for future work could be limiting the neighbor search window for each point to a subset of the available points.
>
> * "Although the evaluation on four categories is reported, Table 2 only gives results for two categories":
> We incorrectly reported four categories, but they are actually three (airplane, chair, sofa). Table 2 reports only two for reasons of space, but figures show all of them and the full table can be found in the appendix.
>
> * "How is the method related to GANs which generates graphs, such as GraphGAN or NetGAN?" :
> Graph generation has the objective of approximating the adjancency matrix, which means predicting the edges, for a class of graphs. The generation of point clouds can be seen as an extension of this problem where every node is associated to a signal, i.e. a vector, such as the x,y,z coordinates or a color, and the network has to learn to reproduce both the adjacency matrix and the signal. A special case for point clouds without color information is that the adjacency matrix is derived from Euclidean neighbors using the signal. This is quite different from predicting arbitrary graphs having no signal on the nodes, as the generative model must learn to produce the x,y,z coordinates and not just predict whether two points should be neighbors.

---

> > ### Comment · AnonReviewer2 · 2018-11-29
> > **Thanks, rating unchanged**
> >
> > Thanks for your detailed response. My rating remains unchanged.

---

### Official Review · AnonReviewer3 · 2018-10-30
**might be an interesting idea; the writing quality is not great; clearly insufficient evaluation**

**Rating:** 6
**Confidence:** 4

**Review:**

The paper proposes a version of GANs specifically designed for generating point clouds. The core contribution of the work is the upsampling operation: in short, it takes as an input N points, and produces N more points (one per input) by applying a graph convolution-like operation.

Pros:
+ The problem of making scalable generative models for point clouds is clearly important, and using local operations in that context makes a lot of sense.

Cons:
- The paper is not particularly well-written, is often hard to follow, and contains a couple of confusing statements (see a non-exhaustive list of remarks below).
- The experimental evaluation seems insufficient: clearly it is possible to come up with more baselines. Even a comparison to other types of generative models would be useful (e.g. variants of VAEs, other types of GANs). There also alternative local graph-convolution-like operations (e.g. tangent convolutions) that are designed for point clouds. In addition, it is quite strange that results are reported not for all the classes in the dataset.

Various remarks:
p.1, “whereby it learns to exploit a self-similarity prior to sample the data distribution”: this is a confusing statement.
p.2, “(GANs) have been shown on images to provide better approximations of the data distribution than other generative models”: This statement is earthier too strong (all other models) or does not say much (some other models)
p.2, “However, this means that they are unable to learn localized features or exploit weight sharing.”: I see the point about no weight sharing in the generator, but feature learning
p.3, “the key difference with the work in this paper is that PointNet and PointNet++ are not
generative models, but are used in supervised problems such as classification or segmentation.”: Yet, the kind of operation that is used in the pointnet++ is quite similar to what you propose?
p.4: “because the high dimensionality of the feature vectors makes the gridding approach unfeasible.”: but you are actually dealing with the point clouds where each point is 3D?

---

> ### Author Response · Authors · 2018-11-14
> **Thank you for your comments**
>
> We thank the reviewer for the comments. We agree that the main strength of the paper is the study of scalable and localized generative models for irregular data like point clouds. However, we want to clarify some aspects of our contribution. In particular, the main focus is on how to use localized operators such as graph convolution when the graph is not known in advance. This problem is specific to GAN generators because the input is a random vector and the 3D point cloud is the output of the network. This is a significant difference with respect to the PointNet and PointNet++ or, in general, supervised problems where the point cloud is the input to the network and therefore it is used to construct the graph. While the actual operations may appear similar as they end up being weighted aggreations of points, the key difference in this work is the non-trivial emergence of localized features, meaning that points that are nearby in the output point cloud are represented in the hidden layers as feature vectors that are "nearby" in this high-dimensional latent space. So, despite the graph not being available as input as in the supervised setting, the training process is able to successively approximate it producing a hierarchy of graph embeddings.
>
> Concerning the experimental evaluation, the focus of the paper is not to provide an extensive comparison of different types of graph convolution or generative models, but rather showing 1) the emergence of localized features; 2) the study of the properties of such features such as being approximate graph embeddings, and the fact that the proposed upsampling operation can exploit self-similarities in the latent representation of the data; 3) the effectiveness of such construction with respect to some baselines for which a straighforward comparison can be made. In this sense the results of comparisons with the baseline models shown in Table 2 (r-GAN-dense and r-GAN-conv) allow to state that the improvement in performance is clearly due to the use of localized operations in the generator and not to some other design choice (e.g. a VAE instead a GAN, or the specific definition of convolution). Only some classes from the Shapenet dataset are reported due to lack of space. These classes are the most commonly used and among the ones with more data samples.
>
> More in detail on the other remarks:
>
> "p.1, “whereby it learns to exploit a self-similarity prior to sample the data distribution”: this is a confusing statement.
> p.2, “(GANs) have been shown on images to provide better approximations of the data distribution than other generative models”: This statement is earthier too strong (all other models) or does not say much (some other models)":
> We improved the wording of these parts in the revised text. The statement about GANs refers to the body of literature on images where it is observed that GANs can generated sharper images than VAEs.
>
> "p.2, “However, this means that they are unable to learn localized features or exploit weight sharing.”: I see the point about no weight sharing in the generator, but feature learning ":
> We are not sure to completely understand this comment (perhaps there is a missing part?). Anyway, the importance of localized features is to exploit a compositionality prior in the data, where the representation of the whole can be constructed from the representations of its parts. Therefore having localized features enables us to have representations of local parts that can be successfully combined using multiple layers.
>
> "p.4: “because the high dimensionality of the feature vectors makes the gridding approach unfeasible.”: but you are actually dealing with the point clouds where each point is 3D?:
> The 3D points are only available at the output of the generator, while hidden layers have a high-dimensional feature vector for each of the points (up to 48 dimensions as shown in Table 1). For this reason the upsampling operation is not trivial and we say that it is infeasible to define a grid over such high dimensional space.

---

> > ### Comment · AnonReviewer3 · 2018-11-22
> > **further comments**
> >
> > Thanks for the elaborate reply, it made things much more clear. I updated the rating accordingly.
> > As for the question about the feature learning, in case we want to use the learned (in unsupervised way) features for prediction tasks (which is often the case), it would probably make more sense to re-use features from the discriminator / encoder, rather than the generator (not to say, that _it is_ interesting that the features learned by the generator have certain properties). In this context the statement that some of the models with non-dense discriminators are unable to learn localised features sounds strange.
> > As for experimental evaluation, exactly because you state that the gains in performance are not supposed to be specific to choices of GANs / VAEs / types of convolutions, it would be interesting to see those across those different model choices.  And you could provide full results in the supplementary material?
> > In general, it seems that authors slightly over-state the differences with / benefits over the existing methods which just makes it more confusing. For example:
> > - there does not seem to be a big practical difference between the _operation_ in PointNet++ and the proposed operation (how it is used, and the entire model are of course still valid contributions)
> > - the comment that generating point clouds is somehow a generalisation of graph generation (a lot of models for graph generation are actually quite capable of also generating the signal in the nodes, e.g. GraphVAE)

---

> > > ### Author Response · Authors · 2018-11-26
> > > **Thank you for your reply**
> > >
> > > We thank the reviewer for the reply and updating the rating.
> > >
> > > Concerning feature learning, we agree that discriminator features can be used in prediction tasks. However, in our statements we always focused on generator features in order to show that they can conveniently exploit localized/hierarchical priors in the data for the generation task. There might have been some confusion about models with non-dense discriminators (or models for supervised problems). Those can indeed learn localized features but they are only used in the discriminator network. In fact, a network with a dense generator and a non-dense discriminator would have useful localized features at the discriminator but a generator that is not capable of exploiting such priors.
> > >
> > > The major difference between the convolution operation used in this work and the aggregations used in PointNet++ is how the contributions of neighbors are taken into account. The Edge-Conditioned convolution used in this paper weighs the neighboring points with a matrix computed from the difference between points. This computation is implemented as a small neural network, which has the advantage of providing "weight sharing" since for the same difference vector the weight to be used in the aggregation is always the same, while if the difference is not exactly the same, the function should be stable (a small perturbation of the difference leads to a small variation in the weights).
> > >
> > > We agree with the reviewer that it would be interesting to test the relative performance of other types of graph convolution or other types of GANs/VAEs (e.g. also using graph convolution at the discriminator). However, there is no space left to include the results in the paper and they would deserve a more detailed analysis than just leaving them in the supplementary material. Also, due to the lack of time to modify the paper we will leave this as future work. Anyway, as stated in the previous response, we believe that the curent experimental results are sufficient to fully measure the advantages of the proposed method and extensive comparisons among different models are outside of the scope of the paper.

---

> > > > ### Comment · AnonReviewer3 · 2018-11-26
> > > > **Re:**
> > > >
> > > > Thanks for the response. I am keeping my rating.

---

### Official Review · AnonReviewer1 · 2018-10-30
**The first work that proposes localized, graph-concolutional GANs for irregular 3D point clouds: fun ideas and exciting to read.**

**Rating:** 9
**Confidence:** 3

**Review:**

This paper proposes graph-convolutional GANs for irregular 3D point clouds that learn domain (the graph structure) and features at the same time. In addition, a method for upsampling at the GAN generator is introduced. The paper is very well written, addresses a relevant problem (classification of 3D point clouds with arbitrary, a priori unknown graph structure) in an original way, and supports the presented ideas with convincing experiments. It aggregates the latest developments in the field, the Wasserstein GAN, edge-conditional convolutions into a concise framework and designs a novel GAN generator. I have only some minor concerns:

1)	My only serious concern is the degree of novelty with respect to (Achlioptas et al., 2017). The discriminator is the same and although the generator is a fully connected network in that paper, it would be good to highlight conceptual improvements as well as quantitative advantages of the paper at hand more thoroughly. Similarly, expanding a bit more on the differences and improvements over (Grover et al., 2018) would improve the paper.

2)	P3, second to last line of 2.1: reference needs to be fixed "…Grover et al. (Grover et al., 2018)"

3)	It would be helpful to highlight the usefulness of artificially generating irregular 3D point clouds from an application perspective, too. While GANs have various applications if applied to images it is not obvious how artificially created irregular 3D point clouds can be useful. Although the theoretical insights presented in the paper are exciting, a more high-level motivation would further improve its quality.

4)	A discussion of shortcomings of the presented method seems missing. While it is understandable that emphasis is put on novelty and its advantages, it would be interesting to see where the authors see room for improvement.

---

> ### Author Response · Authors · 2018-11-14
> **Thank you for your comments**
>
> We thank the reviewer for the comments. With respect to (Achiloptas et al., 2017) our focus is the study of a novel generator architecture that can produce localized features. The main result is that, by estimating a graph from the feature vectors of the hidden layers and aggregating neighbors, the network learns to successively approximate the geometry of the final point clouds. This locality of the representation helps the network to generate higher quality outputs as can be seen by the quantitative results. The reason why the discriminator is kept the same as in (Achiloptas et al., 2017) is to have a fair comparison highlighting that the gain is due to the higher descriptive power of the generator. The work in Grover et al. has a quite different objective, i.e. estimating the adjancency matrix of a graph, and also the technique used is a VAE instead of a GAN.
>
> Points clouds are more and more relevant in practical applications as they can be generated by instruments such as LiDARs or time of flight cameras. Generative models can be useful for many tasks that range from data augmentation to shape completion or inpainting partial data thanks to the features learned by the model. We improved this description in the revised text.
>
> We expanded the discussion on the proposed method. The main drawback is the relatively high complexity of the graph convolution (in part this is due to non-optimized implementations of such operation in the current deep learning frameworks). More work is needed on the upsampling layer since this is important to exploit priors such compositionality or multiresolution scalability. Upsampling also helps reducing the complexity but it is not clear how to perform it in the high dimensional latent spaces while preserving feature locality.

---

### Meta-Review · Area_Chair1 · 2018-12-13
**Strong paper, well received by reviewers -- accept**

**Confidence:** 4
**Recommendation:** Accept (Poster)

**Metareview:**

All reviewers gave an accept rating: 9, 7 &6.
A clear accept -- just not strong enough reviewer support for an oral.